# ERA-SOLVER: ERROR-ROBUST ADAMS SOLVER FOR FAST SAMPLING OF DIFFUSION PROBABILISTIC MODELS

## ABSTRACT

Though denoising diffusion probabilistic models (DDPMs) have achieved remarkable generation results, the low sampling efficiency of DDPMs still limits further applications. Since DDPMs can be formulated as diffusion ordinary differential equations (ODEs), various fast sampling methods can be derived from solving diffusion ODEs. However, we notice that previous fast sampling methods with fixed analytical form are not able to robust with the various error patterns in the noise estimated from pretrained diffusion models. In this work, we construct an error-robust Adams solver (ERA-Solver), which utilizes the implicit Adams numerical method that consists of a predictor and a corrector. Different from the traditional predictor based on explicit Adams methods, we leverage a Lagrange interpolation function as the predictor, which is further enhanced with an error-robust strategy to adaptively select the Lagrange bases with lower errors in the estimated noise. The proposed solver can be directly applied to any pretrained diffusion models, without extra training. Experiments on Cifar10, CelebA, LSUN-Church, and ImageNet 64 × 64 (conditional) datasets demonstrate that our proposed ERA-Solver achieves 3.54, 5.06, 5.02, and 5.11 Frechet Inception Distance (FID) for image generation, with only 10 network evaluations.

## 1 INTRODUCTION

In recent years, denoising diffusion probabilistic models (DDPMs) Ho et al. (2020) have been proven to have potential in data generation tasks such as text-to-image generationPoole et al. (2022); Gu et al. (2022); Kim & Ye (2021); Chen et al. (2022), speech synthesisHuang et al. (2021); Lam et al. (2022); Leng et al. (2022), and 3D generation Poole et al. (2022); Lin et al. (2023); Wang et al. (2023). They build a diffusion process to add noise into the sample and a denoising process to remove noise from the sample gradually. Compared with generative adversarial networks (GANs)Goodfellow et al. (2014) and variational auto-encoders (VAEs)Child (2021), DDPMs have achieved remarkable generation quality. However, due to the properties of the Markov chain, the sampling process requires hundreds or even thousands of denoising steps. Such defects limit the wide applications of diffusion models. Thus, it is an urgent request for a fast sampling of DDPMs.

There have already existed many works for accelerating sampling speed. Some works introduced an extra training stage, such as knowledge distillation method Salimans & Ho (2021); Meng et al. (2022), training sampler Watson et al. (2021), or directly combining with GANs Wang et al. (2022), to obtain a fast sampler. These methods require a cumbersome training process for each task and are black-box samplers due to the lack of theoretical explanations. Denoising diffusion implicit model (DDIM) Song et al. (2020a) and Score-SDESong et al. (2020b) revealed that the sampling can be reformulated as a diffusion ordinary differential equation (ODE) solving process, which inspired many works to design learning-free fast samplers based on numerical methods. PNDM Liu et al. (2021) introduced high order linear multi-step method Małgorzata & Marciniak (2002), which is also called the explicit Adams method, to sample efficiently, with a warming initialization based on Runge-Kutta methods Butcher (1996). DPM-Solver Lu et al. (2022a) and DEIS Zhang & Chen (2022) introduced exponential integrator from ODE literature Atkinson et al. (2011) for efficient sampling. Based on the exponential integrator, DPM-Solver introduced a novel integration variable to derive a fast numerical solver of diffusion ODE, and DEIS utilized the Lagrange interpolation

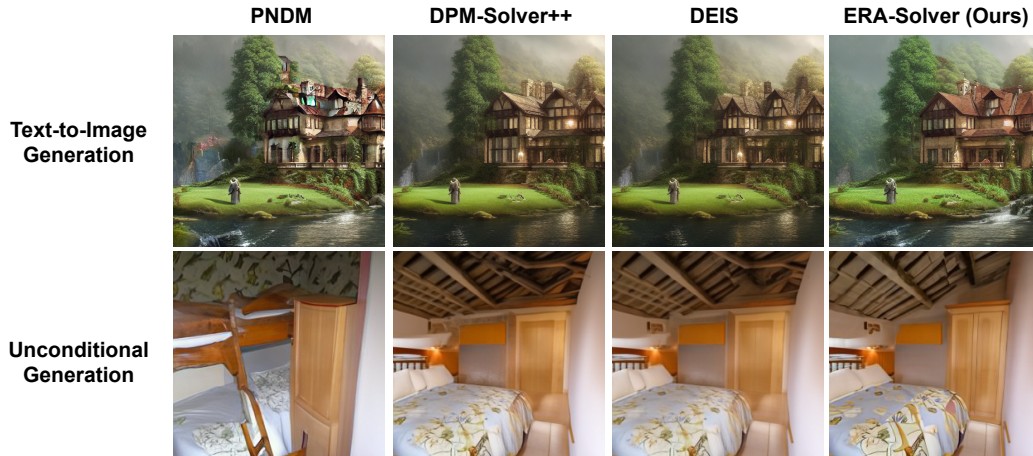

PNDM  DPM-Solver++  DEIS  ERA-Solver (Ours)

Text-to-Image Generation

Unconditional Generation

Figure 1: Generated samples of ERA-Solver and previous fast sampling methods on text-to-image latent diffusion model Rombach et al. (2022) and unconditional pixel-space diffusion model Dhariwal & Nichol (2021).

method. DPM-Solver++ Lu et al. (2022b) is proposed to improve DPM-Solver by combing linear multi-step methods Małgorzata & Marciniak (2002).

The existing fast sampling methods Liu et al. (2021); Lu et al. (2022a); Song et al. (2020a); Zhang & Chen (2022); Lu et al. (2022b); Zhao et al. (2023) all consist of fixed analytical forms to ensure sampling convergence. For example, PNDM Liu et al. (2021) consists of the analytic form with fixed coefficients as formulated in Eq. 7. However, we notice that the noise estimated from the diffusion model is not accurate enough and the error exists at almost every time $t$, especially when time $t$ approaches $0$, as shown in Fig. 2 (b). This phenomenon can be attributed to the training scheme of DDPMs. Furthermore, the trend of the estimation errors varies from the different data manifolds (Fig. 2 (b)). Thus, it limits existing fast sampling methods since the methods with fixed analytical forms can not be robust to various errors from pretrained models and different data manifolds.

In this paper, we aim to design an error-robust numerical solver (as shown in Fig. 2 (a)) of diffusion ODE to speed up the sampling process of DDPMs while achieving good sampling quality. To this end, we focus on implicit Adams solver Małgorzata & Marciniak (2006), a kind of traditional numerical ODE solver, which involves unobserved terms to achieve high-order precision and convergence. In existing ODE literatureAtkinson et al. (2011), predictor-corrector has been introduced to perform implicit Adams solver, which avoids solving the implicit equation. Explicit Adams usually acts as the predictor to predict the unobserved term. However, the traditional predictor-corrector still suffers from the inaccurate estimation of diffusion noise at each sampling step since it is composed of fixed coefficients. Instead of utilizing explicit Adams as the predictor, we adopt the Lagrange interpolation function Sauer & Xu (1995) that interpolates several Lagrange function bases as the predictor. We maintain a buffer of estimated noises observed at previous sampling steps during sampling and adaptively select those estimated noises with low estimation error as the Lagrange function bases, to ensure accurate interpolation results and thus an accurate predictor. In this way, we can obtain a diffusion ODE sampler with not only high convergence (thanks to the implicit Adams methodMałgorzata & Marciniak (2006) and interpolation function) but also good error robustness (thanks to the adaptive selection of the low error Lagrange function bases).

However, it is not easy to select noises with low estimation error as the Lagrange function bases. That is because, unlike the training stage, there exist no reference noises at the sampling stage to judge how accurate the estimated noise is. Thus, we further propose an approach to roughly measure the accuracy of the estimated noise by calculating the difference between the noise obtained by the predictor (as the prediction) and the noise observed at the previous sampling step (as the reference). Based on this measurement, we propose a selection strategy for the buffer that adaptively introduces estimated noises that are more accurate to construct the Lagrange function bases, so as to result in a more accurate predictor, and thus a better ODE solver.

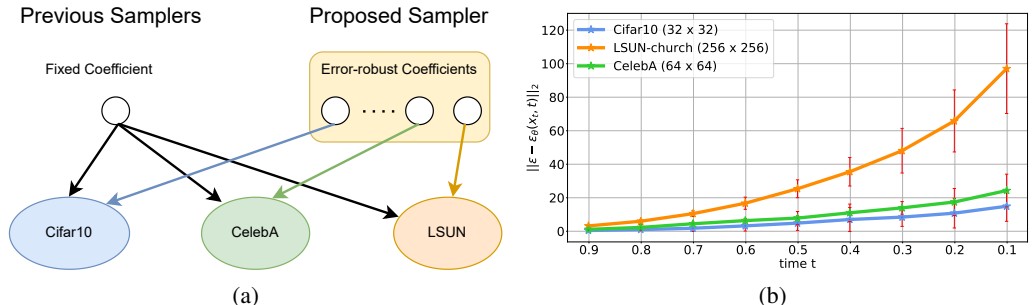

(a)             (b)

Figure 2: (**left**) The main idea of ERA-Solver. ERA-Solver allows flexible sampling coefficients in a unified numerical solver to be error-robust to the various error patterns on different data manifolds. (**right**) The visualization of errors between the estimated noise and ground-truth noise on various data manifolds with the same pretrained diffusion modelSong et al. (2020a). The red bar means the statistical variance.

Our contributions can be summarized as follows:

- We point out the problem that there exist various error patterns of estimated noises when solving diffusion ODEs on different data manifolds and contribute to an error-robust solver (ERA-Solver) that is able to fit the error patterns.

- We explore the potential of the Implicit Adams method Małgorzata & Marciniak (2006). Based on this, we propose the error-robust Lagrange interpolation function that selectively interpolates several Lagrange function bases with lower errors of estimated noises to ensure the ERA-solver is robust to the error in estimated noise.

- Comprehensive experiments on various benchmarks and pretrained diffusion models are conducted to demonstrate the efficiency of ERA-Solver. For instance, with the pretrained model Dhariwal & Nichol (2021) on LSUN-bedroom, ERA-Solver achieves better FID results of 9.09, 7.28, 4.9 compared with the previous best results of 11.01, 9.28, 5.40 with 8, 10, and 20 NFEs.

## 2   PRELIMINARY

We review basic ideas of denoising diffusion probabilistic models (DDPMs), diffusion ordinary differential equations (diffusion ODEs), and the existing training-free numerical methods for fast sampling.

### 2.1   DENOISING DIFFUSION PROBABILISTIC MODELS

To sample from a complex data distribution $q(x_0)$, denoising diffusion probabilistic models (DDPMs) Ho et al. (2020) introduce a forward diffusion process to gradually add noise to data and a parameterized network $\theta$ to predict the noise hidden in the noisy data $x_t$.

**Forward diffusion process.** The diffusion process is modeled as a transition distribution:

$$q(\mathbf{x}_t|\mathbf{x}_{t-1}) := \mathcal{N}(\mathbf{x}_t; \sqrt{\alpha_t}\mathbf{x}_{t-1}, (1-\alpha_t)\mathbf{I}), \tag{1}$$

where $\alpha_1, ..., \alpha_T$ are fixed parameters. With the transition distribution above, noisy distribution conditioned on clean data $x_0$ can be formulated as follows:

$$q(\mathbf{x}_t|\mathbf{x}_0) = \mathcal{N}(\mathbf{x}_t; \sqrt{\bar{\alpha}_t}\mathbf{x}_0, (1-\bar{\alpha}_t)\mathbf{I}), \tag{2}$$

where $\bar{\alpha}_t = \prod_{s=1}^{t} \alpha_s$. When $t$ is large enough, the Markov process will converge to a Gaussian steady-state distribution $\mathcal{N}(0, I)$.

**Training scheme.** With the parameterized noise estimation $\epsilon_\theta$, the training objective of $\theta$ can be written as following:

$$L_{t-1} = \mathbb{E}_{\mathbf{x}_0, \boldsymbol{\epsilon}}[\omega(t)||\boldsymbol{\epsilon} - \boldsymbol{\epsilon}_\theta(\mathbf{x}_t, t)||^2]. \tag{3}$$

where $\mathbf{x}_0 \sim q(x_0)$, $\epsilon \sim \mathcal{N}(0, I)$, and $\omega(t)$ is a weight function that consists of the $\alpha(t)$.

Though DDPMs have good theoretical properties, they suffer from sampling efficiency. It usually requires hundreds of network evaluations, which limits various downstream applications for DDPMs. There already existed methods Watson et al. (2021); Lyu et al. (2022); Lam et al. (2022); Salimans & Ho (2021); Meng et al. (2022) which depend on extra training stage to derive fast sampling methods. The training-based methods usually require tremendous training costs for different data manifolds and tasks, which inspires many works Song et al. (2020b;a); Liu et al. (2021); Lu et al. (2022a;b); Zhang & Chen (2022) to explore a training-free sampler based on numerical methods.

## 2.2 NUMERICAL METHODS FOR FAST SAMPLING

Score-SDE Song et al. (2020b) demonstrated that the reverse denoising process of DDPMs can be formulated as follows:

$$\mathbf{x}_{t_{i+1}} = \sqrt{\bar{\alpha}_{t_{i+1}}}\Big(\frac{\mathbf{x}_t - \sqrt{1 - \bar{\alpha}_{t_i}}\boldsymbol{\epsilon}_\theta(\mathbf{x}_{t_i}, t_i)}{\sqrt{\bar{\alpha}_{t_i}}}\Big) + \sqrt{1 - \bar{\alpha}_{t_{i+1}} - \sigma_{t_i}^2}\boldsymbol{\epsilon}_\theta(\mathbf{x}_{t_i}, t_i) + \sigma_{t_i}\mathbf{z}, \tag{4}$$

where $\{t_i\}_{i=0}^N$ is the iteration time steps we introduced for the ease of description and $z$ is the random Gaussian noise. $t_0$ represents the beginning time and $t_N$ represents the end time. When $\sigma$ equals 0, the reverse process can be reformulated as a diffusion ODE Liu et al. (2021); Song et al. (2020a):

$$\frac{\mathrm{d}\mathbf{x}}{\mathrm{d}t} = -\bar{\alpha}'(t)\Big(\frac{x(t)}{2\bar{\alpha}(t)} - \frac{\boldsymbol{\epsilon}_\theta(\mathbf{x}_t, t)}{2\bar{\alpha}(t)\sqrt{1 - \bar{\alpha}(t)}}\Big), \tag{5}$$

where $\alpha(t)$ is the continuous version of $\{\bar{\alpha}_{t_i}\}_{i=1}^N$. Denoising diffusion implicit model (DDIM) Song et al. (2020a) provided a discrete form for solving the diffusion ODE above (Eq.31). Its sampling scheme can be formulated as:

$$\mathbf{x}_{t_{i+1}} = \frac{\sqrt{\bar{\alpha}_{t_{i+1}}}}{\sqrt{\bar{\alpha}_{t_i}}}\mathbf{x}_{t_i} + \Big(\sqrt{1 - \bar{\alpha}_{t_{i+1}}} - \frac{\sqrt{\bar{\alpha}_{t_{i+1}}(1 - \bar{\alpha}_{t_i})}}{\sqrt{\bar{\alpha}_{t_i}}}\Big)\boldsymbol{\epsilon}_{t_i}, \tag{6}$$

where $\boldsymbol{\epsilon}_{t_i} = \boldsymbol{\epsilon}_\theta(x_{t_i}, t_i)$. Such a property inspires many works to introduce numerical methods, such as Runge-Kutta Butcher (1996) method and linear multi-step method Wells (1982), to construct efficient solvers. PNDM Liu et al. (2021) combined DDIM and explicit Adams Małgorzata & Marciniak (2002), a kind of linear multi-step method, to derive a novel numerical solver of diffusion ODE. The $\boldsymbol{\epsilon}_{t_i}$ in Eq. 6 is reformulated as following:

$$\boldsymbol{\epsilon}_{t_i} = \frac{1}{24}(55\boldsymbol{\epsilon}_\theta(\mathbf{x}_{t_i}, t_i) - 59\boldsymbol{\epsilon}_\theta(\mathbf{x}_{t_{i-1}}, t_{i-1}) + 37\boldsymbol{\epsilon}_\theta(\mathbf{x}_{t_{i-2}}, t_{i-2}) - 9\boldsymbol{\epsilon}_\theta(\mathbf{x}_{t_{i-3}}, t_{i-3})). \tag{7}$$

Different from PNDM, DPM-SolverLu et al. (2022a) applied the exponential integrator from ODE literature Atkinson et al. (2011) and proposed a novel integration variable so as to conveniently approximate integration, deriving a fast numerical solver with explicit analytical form. It is further improved in DPM-Solver++ Lu et al. (2022b) by combining linear multi-step method Atkinson et al. (2011). DEIS Zhang & Chen (2022) proposed to utilize the Lagrange interpolation function to estimate the $\epsilon_\theta(x_t, t)$. UniPC Zhao et al. (2023) proposed a unified predictor-corrector framework for fast sampling. More discussions can be found in Appendix. A.

Different from previous methods, we are motivated by the inaccurately estimated noises of pretrained diffusion models across most sampling time, especially when time $t_i$ is close to 0. Furthermore, the error patterns may vary between different data manifolds, as shown in Fig.2 (b). Previous numerical methods with fixed analytical forms can not fit the various error patterns generally. In this paper, we propose an error-robust solver based on implicit Adams numerical methods. It consists of the Lagrange interpolation function with a novel selection strategy to select noises with low estimation error to construct the interpolation function adaptively.

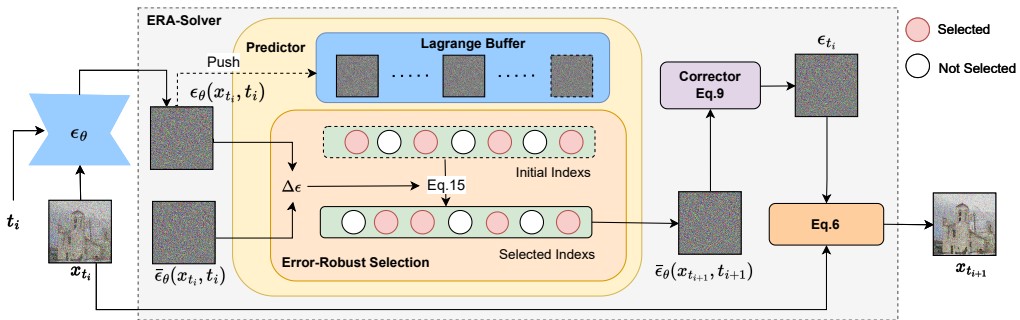

Figure 3: The pipeline of ERA-Solver. The sampling scheme is based on the predictor-corrector method for implicit Adams. Our predictor is robust to the errors of the estimated noises from pretrained models. The sampling starts from normal Gaussian noise $x_{t_0}$ and performs a denoising scheme (from $x_{t_i}$ to $x_{t_{i+1}}$) iteratively to get the final generated image.

## 3 ERA-SOLVER

In this section, we first point out that the error in the estimated noise $\boldsymbol{\epsilon}_\theta(\mathbf{x}_{t_i}, t_i)$ by the network $\theta$ limits the previous numerical fast samplers and introduce implicit Adams numerical solver (Sec. 3.1). Then, we apply predictor-corrector sampling and leverage a Lagrange interpolation function as the predictor (Sec. 3.2). We design an error distance to measure the accuracy of the estimated noise and enhance the proposed predictor with an error-robust strategy to adaptively select the Lagrange bases with lower noise estimation error (Sec. 3.3). The whole sampling process is shown in Fig. 3.

### 3.1 IMPLICIT ADAMS METHODS

The sampling of DDPMs starts from a prior noise distribution $\mathbf{x}_{t_0} \sim \mathcal{N}(0, I)$, and iteratively denoises $\mathbf{x}_{t_i}$ to $\mathbf{x}_{t_{i+1}}$ until time $t$ reaches $0$. In the sampling process, the most time-consuming step is network evaluation. Assuming we have a pretrained noise estimation model $\boldsymbol{\epsilon}_\theta$, our goal is to achieve good generation quality with as few evaluation times as possible.

We notice that the noise estimation error exists across almost every sampling time, especially when time $t_i$ is close to $0$. The changing trend of estimation errors may also vary between different training manifolds, as shown in Fig. 1. Such a property can be attributed to the training scheme (Eq. 3) of DDPMs. It limits previous numerical high-order solvers Liu et al. (2021); Lu et al. (2022a); Song et al. (2020a); Zhang & Chen (2022) since they are all based on the fixed analytical form to reach the sampling convergence for solving the diffusion ODE efficiently. The solver with a fixed analytical form can not be able to fit the various estimation errors from different sampling time and data manifolds. Thus, they may suffer from obvious estimation errors, which motivates us to explore the error-robust numerical solver.

In this paper, we first explore the potential of implicit Adams solver Małgorzata & Marciniak (2006). Different from explicit Adams (Eq. 7), implicit Adams involves the unobserved noise term, and the $\boldsymbol{\epsilon}_{t_i}$ in Eq. 31 is reformulated as follows:

$$\boldsymbol{\epsilon}_{t_i} = \frac{1}{24}(9\boldsymbol{\epsilon}_\theta(\mathbf{x}_{t_{i+1}}, t_{i+1}) + 19\boldsymbol{\epsilon}_\theta(\mathbf{x}_{t_i}, t_i) - 5\boldsymbol{\epsilon}_\theta(\mathbf{x}_{t_{i-1}}, t_{i-1}) + \boldsymbol{\epsilon}_\theta(\mathbf{x}_{t_{i-2}}, t_{i-2})). \tag{8}$$

It shares the same convergence order with explicit Adams but has better stability Atkinson et al. (2011). It can be noticed that $\mathbf{x}_{t_{i+1}}$ can be observed only when $\boldsymbol{\epsilon}_{t_i}$ is achieved, while the $\boldsymbol{\epsilon}_{t_i}$ contains unobserved term $\boldsymbol{\epsilon}_\theta(\mathbf{x}_{t_{i+1}}, t_{i+1})$, which makes it challenging to solve implicit equations and may need more time-consuming iteration steps. This greatly limits the implicit Adams method to be a fast solver for diffusion ODEs.

Fortunately, in numerical ODE literature, the sampling efficiency of implicit Adams can be improved with a predictor-corrector sampling scheme Diethelm et al. (2002). Specifically, the predictor makes a rough estimation of unobserved term $\bar{\boldsymbol{\epsilon}}_\theta(\mathbf{x}_{t_{i+1}}, t_{i+1})$ and the corrector derives the precise $\mathbf{x}_{t_{i+1}}$, which can reformulate Eq. 8 as follows:

$$\boldsymbol{\epsilon}_{t_i} = \frac{1}{24}(9\bar{\boldsymbol{\epsilon}}_\theta(\mathbf{x}_{t_{i+1}}, t_{i+1}) + 19\boldsymbol{\epsilon}_\theta(\mathbf{x}_{t_i}, t_i) - 5\boldsymbol{\epsilon}_\theta(\mathbf{x}_{t_{i-1}}, t_{i-1}) + \boldsymbol{\epsilon}_\theta(\mathbf{x}_{t_{i-2}}, t_{i-2})). \tag{9}$$

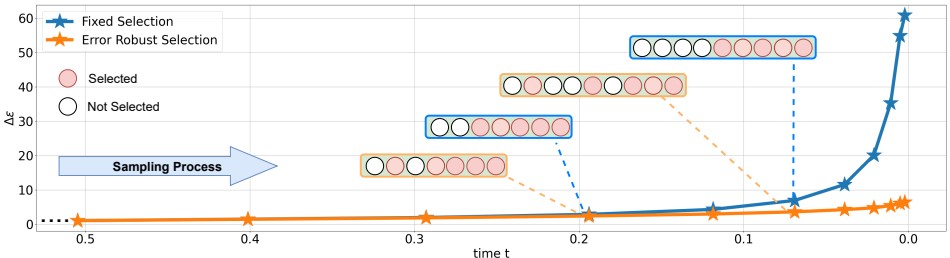

Figure 4: $\Delta\epsilon$ comparison of the error-robust selection process and fixed selection process. $\Delta\epsilon$ is calculated based on Eq. 13 instead of the training loss in Eq. 3 on Cifar10 Krizhevsky et al. (2009). The sampling NFE is set to 20 and $k$ is set to 5.

The traditional predictor-corrector utilizes explicit Adams (Eq. 7) to perform predictor to make $\mathbf{x}_{t_{i+1}}$ observed so as to derive $\bar{\epsilon}_\theta(\mathbf{x}_{t_{i+1}}, t_{i+1})$. However, it still consists of the fixed analytical form and can not fit the various estimation errors at the training stage (Fig. 2).

## 3.2 Predictor with Lagrange Interpolation Function

In this paper, we propose to utilize noises observed at previous sampling steps and construct the Lagrange interpolation function as the predictor to predict unobserved term $\epsilon_\theta(\mathbf{x}_{t_{i+1}}, t_{i+1})$. In this way, we can design an adaptive strategy to select Lagrange function bases to construct the error-robust predictor.

Specifically, we maintain a buffer about all previously estimated noises, which have been observed and need no extra computations, and its corresponding time:

$$\{(t_n, \epsilon_\theta(\mathbf{x}_{t_n}, t_n)), n = 0, 1, .., i\}. \tag{10}$$

The maintained buffer is also called the Lagrange buffer in this paper. Assume that the interpolation order is $k$, the selected function bases to construct the Lagrange function can be written as $\{(t_{\tau_m}, \epsilon_\theta(\mathbf{x}_{t_{\tau_m}}, t_{\tau_m})), m = 0, ...k - 1\}$. The corresponding Lagrange interpolation function can be formulated as:

$$l_m(t) = \prod_{l=0, l\neq m}^{k-1} (\frac{t - t_{\tau_l}}{t_{\tau_m} - t_{\tau_l}}),$$

$$L_\epsilon(t) = \sum_{m=0}^{k-1} l_m(t) * \epsilon_\theta(\mathbf{x}_{t_{\tau_m}}, t_{\tau_m}), \tag{11}$$

where $\tau_l$ belongs to the maintained Lagrange buffer and has already been observed. At time $t_{i+1}$, we can derive an estimation about $\epsilon_\theta(\mathbf{x}_{t_{i+1}}, t_{i+1})$:

$$\bar{\epsilon}_\theta(\mathbf{x}_{t_{i+1}}, t_{i+1}) = L_\epsilon(t_{i+1}). \tag{12}$$

With this prediction, we apply the corrector process in Eq. 9 to get the $\epsilon_{t_i}$ and Eq. 6 to get the denoised sample $\mathbf{x}_{t_{i+1}}$.

It can be noticed that the proposed predictor makes use of the observed noise estimations and involves no network evaluations. Furthermore, the Lagrange bases in the predictor can be adaptively selected from those noise estimations with low errors, which is more error-robust. We introduce this error-robust selection strategy in the next subsection.

## 3.3 Error-Robust Selection Strategy

In this part, our goal is to design an error-robust selection strategy for the maintained Lagrange buffer. When the interpolation order is $k$, the intuitive selection approach is to make a fixed selection of the last $k$ estimated noises from the maintained Lagrange buffer, which means $\tau_m = i - m$ in Eq. 11. However, we notice that the noise estimation error tends to increase as time $t_i$ approaches 0 (Fig. 2), in which case the fixed selection strategy may aggregate the noise estimation errors from Lagrange

---

**Algorithm 1** ERA-Solver

---
1: **Input**: $\{t_i\}_{i=0}^N, k, \epsilon_\theta$
2: **Instantiate**: $\mathbf{x}_{t_0} \sim \mathcal{N}(0, \mathbf{I})$, buffer $\Omega = \varnothing$, $\Delta\epsilon = \lambda$
3: $\Omega = \Omega \cup \{(t_0, \epsilon_\theta(\mathbf{x}_{t_0}, t_0))\}$
4: **for** $i$ in $0, 1, \cdots, N-1$ **do**
5:     **if** $i < k-1$ **then**
6:         Derive $x_{t_{i+1}}$ based on Eq. 6 and $\epsilon_\theta(\mathbf{x}_{t_i}, t_i)$
7:         $\Omega = \Omega \cup \{(t_{i+1}, \epsilon_\theta(\mathbf{x}_{t_{i+1}}, t_{i+1}))\}$
8:     **else**
9:         Calculate $\{\bar{\tau}_m\}_{m=0}^{k-1}$ via Eq. 14
10:        Calculate $\{\tau_m\}_{m=0}^{k-1}$ via Eq. 15 and $\Delta\epsilon$
11:        Derive Lagrange function $L_\epsilon$ via Eq. 11 and $\tau_m$
12:        $\bar{\epsilon}_\theta(\mathbf{x}_{t_{i+1}}, t_{i+1}) \leftarrow L_\epsilon(t_{i+1})$
13:        Calculate $\epsilon_{t_i}$ via $\Omega$, $\bar{\epsilon}_\theta(\mathbf{x}_{t_{i+1}}, t_{i+1})$, and Eq. 9
14:        Derive $\mathbf{x}_{t_{i+1}}$ based on Eq. 6 and $\epsilon_{t_i}$
15:        $\Omega = \Omega \cup \{(t_{i+1}, \epsilon_\theta(\mathbf{x}_{t_{i+1}}, t_{i+1}))\}$
16:        Update $\Delta\epsilon$ via Eq. 13 and $\bar{\epsilon}_\theta(\mathbf{x}_{t_{i+1}}, t_{i+1})$
17:     **end if**
18: **end for**
19: **return** $\mathbf{x}_{t_N}$

---

buffer and make the constructed Lagrange function inaccurate for prediction at time $t_{i+1}$. It motivates us to seek a reasonable measure of the error in estimated noise so as to select those noises with low estimation error for Lagrange interpolation.

**Error Measure for Estimated Noise.** Since there exists no ground-truth noise in the sampling process, it is hard to measure the error in estimated noise. To this end, we propose to utilize the observed noise term $\epsilon_\theta(\mathbf{x}_{t_i}, t_i)$ as the target noise and the predicted noise term $\bar{\epsilon}_\theta(\mathbf{x}_{t_i}, t_i)$ from last sampling step as the estimated noise to calculate the approximation error, which can be seen in Fig. 3. It can be formulated as follows:

$$\Delta\epsilon = ||\epsilon_\theta(\mathbf{x}_{t_i}, t_i) - \bar{\epsilon}_\theta(\mathbf{x}_{t_i}, t_i)||_2. \tag{13}$$

$\epsilon_\theta(\mathbf{x}_{t_i}, t_i)$ is observed based on $x_{t_i}$, which is achieved via the $\bar{\epsilon}_\theta(\mathbf{x}_{t_i}, t_i)$ and Eq. 31. When the error of estimated noise from pretrained models increases, it tends to be hard for $\bar{\epsilon}_\theta(\mathbf{x}_{t_i}, t_i)$ to approximate $\epsilon_\theta(\mathbf{x}_{t_i}, t_i)$. As shown in Fig.4, our error measure in the sampling process shares a similar trend as the error of estimated noises in the training process (Fig. 2), which demonstrates the rationality of the proposed error measure. More analysis can be found in Appendix. B.2.

**Selection Strategy.** The high-level idea of our error-robust selection strategy is that we tend to choose those estimated noises from the Lagrange buffer with low errors measured by Eq. 13 as the Lagrange bases.

The selection strategy should balance the interpolation accuracy of $\bar{\epsilon}_\theta(x_{t_{i+1}}, t_{i+1})$ and the error robustness. If the selected function bases are all gathered at the beginning of the buffer, the accuracy of $\bar{\epsilon}_\theta(x_{t_{i+1}}, t_{i+1})$ will be compromised since the selected function bases are too far from the currently estimated noise. Next, we try to utilize power function to build the selection process.

When sampling at time $t_i$, the length of the Lagrange buffer is $i+1$. We initialize $k$ indexes uniformly to cover the whole buffer:

$$\widehat{\tau}_m = (i/k) * m, m = 1, 2, .., k \tag{14}$$

Then we utilize the power function as an index translator. We parameterize the power function with the error measure (Eq. 13) so that the translation of initial indexes can be formulated as:

$$\tau_m = \lfloor (\widehat{\tau}_m/i)^{\Delta\epsilon/\lambda} * i \rfloor. \tag{15}$$

where $\lambda$ is a hyperparameter to adjust the scale. Theorem 1 ensures that the proposed selection method can be robust to the changing trend of the estimation error. Furthermore, $\lambda$ can be regarded as the manifold-aware hyperparameter. We can adjust $\lambda$ to make ERA-Solver fit the different estimation error patterns. More evidence can be found in the experiment part.

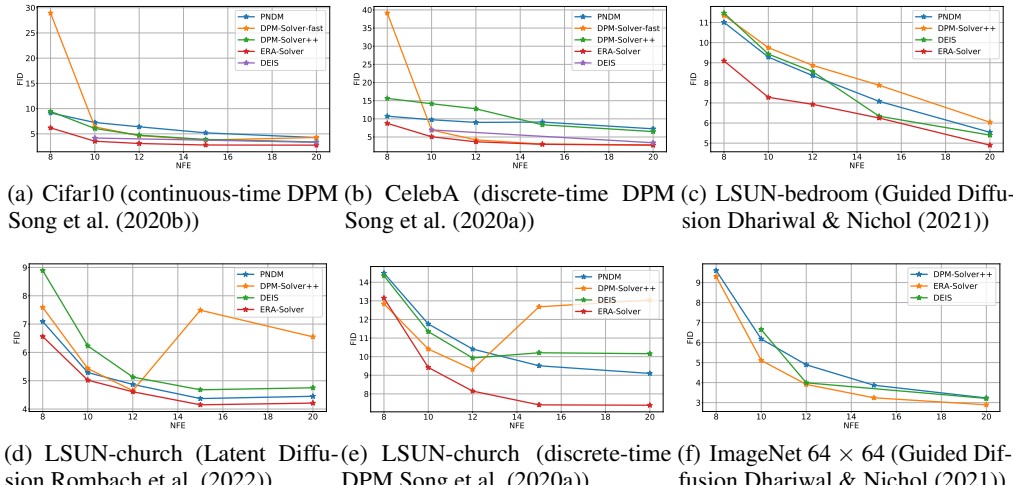

(a) Cifar10 (continuous-time DPM Song et al. (2020b))

(b) CelebA (discrete-time DPM Song et al. (2020a))

(c) LSUN-bedroom (Guided Diffusion Dhariwal & Nichol (2021))

(d) LSUN-church (Latent Diffusion Rombach et al. (2022))

(e) LSUN-church (discrete-time DPM Song et al. (2020a))

(f) ImageNet 64 × 64 (Guided Diffusion Dhariwal & Nichol (2021))

Figure 5: Generation quality measured by FID ↓ on various datasets and pretrained DPMs, varying the number of function evaluation (NFE).

As shown in Fig. 4, our selection strategy introduces more indexes near the beginning of the Lagrange buffer when the error of estimated noises increases, while maintaining the most indexes near the currently estimated point to achieve the error robustness. In this way, our selection strategy of buffer makes our Lagrange interpolation function more accurate in an adaptive way (Fig. 4), which contributes to an error-robust Adams solver.

## 3.4 OVERALL SAMPLING ALGORITHM

In this part, we summarize our sampling algorithm. Given a pretrained diffusion model $\epsilon_\theta$, we sample from a prior noise $x_{t_0} \sim \mathcal{N}(0, I)$ and iteratively denoise from $x_{t_i}$ to $x_{t_{i+1}}$ until time $t_i$ reaches 0 and $x_{t_N}$ is our final generated sample. The sampling algorithm is based on the predict-corrector method. The $k$ represents the Lagrange interpolation order. For the initialization of the Lagrange buffer, the first $k$ sampling steps are based on DDIM Song et al. (2020a) sampling scheme. The details of the sampling process can be found in Alg. 1.

## 3.5 THEORETICAL ANALYSIS

**Theorem 1.** *When $k \geq 3$, ERA-Solver has a third-order local approximation error and a second-order convergence.*

**Theorem 2.** *When the error measure $\Delta\epsilon$ is large enough, the selected indexes are gathered at the beginning, and vice versa.*

Theorem 1 ensures that ERA-Solver is an efficient numerical solver. Theorem 2 ensures that the proposed selection function contributes to an error-robust selection strategy, making ERA-Solver achieve better generation quality. Detailed proofs of the theories above are provided in Appendix. C.

## 4 EXPERIMENT

In this section, we demonstrate that, as a training-free sampler, ERA-Solver is able to speed up the sampling of pretrained diffusion models greatly. The error-robust property helps ERA-Solver achieve better generation quality on various datasets. We adopt Frechet Inception Distance (FID) Heusel et al. (2017) as the metric to evaluate the generation quality of all sampling methods. All experiment results are evaluated based on $50k$ generated samples.

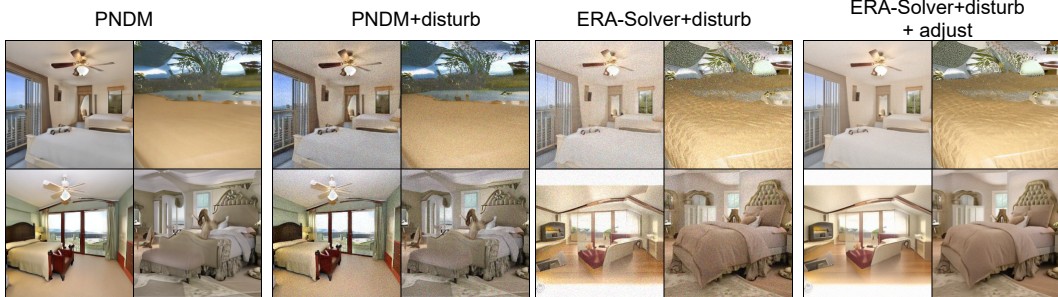

Figure 6: The demonstration of how ERA-Solver helps improve sampling quality. PNDM and ERA-Solver share the same numerical method (Adams method) and the same convergence order. However, ERA-Solver can be robust to the designed disturbing score by adjusting $\lambda$ manually.

## 4.1 How ERA-Solver Works to Improve Sampling Quality

In this subsection, we discuss how ERA-Solver improves sampling quality. Different from previous sampling methods, ERA-Solver takes the estimation error in the pretrained DPMs into consideration and introduces ERS to be robust to the errors. Since the estimation error distribution in the pretrained diffusion model is intractable, we manually design a disturbance score:

$$\boldsymbol{\epsilon}'_{t\theta}(x_{t_i}, t_i) = \boldsymbol{\epsilon}_{t\theta}(x_{t_i}, t_i) + 0.01 * (1.0 - t_i) * \boldsymbol{\epsilon}, \boldsymbol{\epsilon} \sim \mathcal{N}(0, I) \tag{16}$$

Such a disturbing noise , which tends to be strong when time t is small, acts as a simple simulation of estimation error patterns (Fig. 2(b)). In the next, we perform sampling with the disturbing score and the original score separately. We select PNDM Liu et al. (2021) to make a comparison with ERA-Solver since PNDM and ERA-Solver are all based on Adams numerical methods (ERA-Solver is based on implicit Adams, while PNDM is based on explicit Adams). We select LSUN-bedroom as the benchmark and pretrained Guided Diffusion Dhariwal & Nichol (2021) for sampling. The sampling results can be seen in Fig. 6, from which we can observe that PNDM can be affected by the disturbing score, while ERA-Solver can alleviate the effect by adjusting $\lambda$. FID evaluation results can be found in Table. 1 of Appendix.

The estimation error in the pretrained diffusion model can be regarded as an intractable disturbing pattern, which means when adjusting $\lambda$ in ERA-Solver, ERA-Solver is seeking the best sampling coefficients to fit this error pattern.

## 4.2 Comparison with Previous Fast Sampling Methods

The comparison results on various datasets and pretrained diffuision models are shown in Fig. 5. It can be observed that ERA-Solver achieves FID improvement by an obvious margin at most NFEs. The best hyperparameter $\lambda$ varies between datasets and pretrained DPMs. For the sampling of ERA-Solver in Fig. 5, we set $\lambda = 10.0, 30.0, 1.0, 20.0, 8.0, 20.0$ in order. The different best hyperparameters for various datasets verify our analysis in Fig. 2 and Sect. 3.1 to some extent. More details are provided in the Appendix. D.

## 4.3 Limitation

Although ERA-Solver helps improve the sampling quality, it still has limitations. Since the error-robust Lagrange interpolation method will consider all previously estimated noises, the maintained buffer will be long. Thus, the necessary computation time will be slightly more than other methods. We provide the detailed computation time in the Appendix. E.4.

## 5 Conclusion

In this paper, we propose an error-robust Adams solver (ERA-Solver) that consists of a predictor and a corrector. We leverage the Lagrange interpolation function to perform the predictor and further propose an error measure for the sampling process and an error-robust strategy to enhance the predictor. Experiments demonstrate that ERA-Solver achieves better generation quality on various datasets and pretrained diffusion models at few NFEs.

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

## A  APPENDIX

## B  METHOD DISCUSSION

### B.1  RELATIONSHIPS WITH PREVIOUS FAST SAMPLERS

In this subsection, we highlight the contribution of ERA-Solver. The main difference between ERA-Solver and previous numerical samplers is the perspective to improve the sampling quality. We are motivated by the various estimation error patterns hidden in the pretrained DPMs. The fixed sampling scheme may not fit such various error patterns caused by different data manifolds and network architectures. The error may also limit the previous numerical solvers which depends on the assumptions that $\epsilon_\theta(x_t, t) = \nabla_{x_t} \log p(x_t)$. ERA-Solver contributes to a flexible numerical sampling framework that enables the adjustable sampling coefficients and high-order sampling convergence (Theorem. C.1) simultaneously. Based on such a framework, the designed Error-Robust Selection strategy can make ERA-Solver be robust to the various error patterns by adjusting its $\lambda$.

The most silimiar work with ERA-Solver is PNDM Liu et al. (2021). PNDM and ERA-Solver are both mainly based on Adams numerical method. However, different from PNDM which involves Explicit Adams, ERA-Solver utilizes the Implicit Adams to perform effect sampling. Furthermore, ERA-Solver supports constructing error-robust sampling coefficients while PNDM can not. (Fig. 6). We provide the FID evaluation results in the Table. 1.

Table 1: FID Results on LSUN-bedroom Yu et al. (2015) with pretrained Guided Diffusion model Dhariwal & Nichol (2021). The methods are evaluated with 5000 generated samples. The 'disturb' means the estimated noise in the sampling process is rewritten as Eq. 16.

| Sampling method \ NFE | 10 | 12 | 15 |
|---|---|---|---|
| PNDM | 12.58 | 11.73 | 9.48 |
| PNDM + disturb | 50.42 | 44.08 | 34.89 |
| ERA-Solver | 10.93 | 10.09 | 9.18 |
| ERA-Solver + disturb | 45.22 | 38.27 | 14.58 |
| ERA-Solver + disturb + adjust $\lambda$ | 12.06 (-33.16) | 11.14 (-27.13) | 9.40 (-5.18) |

DEIS Zhang & Chen (2022) proposed to utilize the Exponential Integrator (EI) to simplify the integral process in diffusion ODEs and introduced the Lagrange interpolation function to approximate the integral term. ERA-Solver also utilizes the Lagrange interpolation function. However, DEIS still can not fit the various error patterns since its selection strategy of Lagrange bases is fixed. Different from DEIS, the robustness of ERA-Solver to estimation errors mainly comes from the designed selection strategy of Lagrange bases.

UniPC Zhao et al. (2023) proposed a unified predictor-corrector sampling framework that supports higher-order sampling convergence. However, limited by the various estimation errors in the pre-trained DPMs, the higher-order sampling numerical convergence may not bring more sampling quality. ERA-Solver also involves predictor-corrector methods. However, the main body of the predictor-corrector in ERA-Solver is the estimated noise. We compare ERA-Solver and UniPC in the Table. 5.

### B.2 RATIONALTIY OF ERROR MEASURE

The estimation error can not be observed in the sampling process, which limits us to designing an error-robust sampling scheme. To alleviate this issue, we propose an error measure $\Delta\epsilon = \bar{\epsilon}_\theta(x_t, t) - \bar{\epsilon}_\theta(x_t, t)$, where $\bar{\epsilon}_\theta(x_t, t)$ is derived by constructing Lagrange interpolation function. Intuitively, if the currently estimated noise has large errors, the constructed Lagrange interpolation which consists of the nearby function points is likely shares the same error. Thus, the approximation of the Lagrange interpolation function is likely to be more inaccurate. In this way, the proposed error measure $\Delta\epsilon$ can be sensitive to the estimation errors from the training scheme. Furthermore, the hyperparameter $\lambda$ can be seen as the critical value of $\Delta\epsilon$ to control the aggregation at the beginning of the buffer. The detailed analysis can be found in Theorem. C.2.

## C THEORY ANALYSIS

### C.1 PROOF OF THEOREM 1

To analyze the convergence order of ERA-Solver, we first need to calculate its discretization error of the general form. For ease of description, we rewritte $\mathbf{x}_{t_i}$ and $\bar{\alpha}_{t_i}$ as the continuous form $x(t)$ and $\alpha(t)$. We rewritte $\epsilon_\theta(x(t), t)$ as $\epsilon_\theta(t)$. Firstly we recall the diffusion ODE Song et al. (2020b); Liu et al. (2021):

$$\frac{\mathrm{d}\mathbf{x}}{\mathrm{d}t} = -\bar{\alpha}'(t)\left(\frac{x(t)}{2\bar{\alpha}(t)} - \frac{\epsilon_\theta(t)}{2\bar{\alpha}(t)\sqrt{1 - \bar{\alpha}(t)}}\right). \tag{17}$$

To solve this ODE, we utilize integral on both sides and can get the general solver as follows:

$$\mathbf{x}(t + \delta) - \mathbf{x}(t) = -\int_t^{t+\delta} \alpha'(\tau)\left(\frac{x(\tau)}{2\bar{\alpha}(\tau)} - \frac{\epsilon_\theta(\tau)}{2\bar{\alpha}(\tau)\sqrt{1 - \bar{\alpha}(\tau)}}\right)\mathrm{d}\tau. \tag{18}$$

Following DEIS Zhang & Chen (2022) and DPM-Solver Lu et al. (2022a), to eliminate the integration error caused by linear terms, we apply the Exponential Integrator (EI) so that the general solver can be rewritten as:

$$\mathbf{x}(t + \delta) = e^{\int_t^{t+\delta} f(\tau)\mathrm{d}\tau}\mathbf{x}(t) + \int_t^{t+\delta} G(\tau)\epsilon_\theta(\tau)\mathrm{d}\tau, \tag{19}$$

where $G(\tau) = e^{\int_\tau^{t+\delta} f(\tau)\mathrm{d}\tau}\frac{\alpha'(\tau)}{2\alpha(\tau)\sqrt{1-\alpha(\tau)}}$. Next, we introduce an intermediate numerical solution $\bar{\mathbf{x}}(t + \delta)$ and it reads:

$$\bar{\mathbf{x}}(t + \delta) = e^{\int_t^{t+\delta} f(\tau)\mathrm{d}\tau}\mathbf{x}(t) + \int_t^{t+\delta} G(t)\frac{1}{\delta}\epsilon_\theta(\tau)\mathrm{d}\tau$$
$$= e^{\int_t^{t+\delta} f(\tau)\mathrm{d}\tau}\mathbf{x}(t) + G(t)\int_t^{t+\delta} \frac{1}{\delta}\epsilon_\theta(\tau)\mathrm{d}\tau. \tag{20}$$

Then, we introduce the numerical solution $\mathbf{x}_{pndm}(t + \delta)$ in PNDM Liu et al. (2021). It can be written as:

$$\mathbf{x}_{pndm}(t + \delta) = e^{\int_t^{t+\delta} f(\tau)\mathrm{d}\tau}\mathbf{x}(t) + G(t)\frac{1}{24}(55\epsilon_\theta(t) - 59\epsilon_\theta(t - \delta) + 37\epsilon_\theta(t - 2\delta) - 9\epsilon_\theta(t - 3\delta)). \tag{21}$$

PNDM Liu et al. (2021) has proven that it has a 3-order local approximation error. Thus, we have:

$$|\mathbf{x}(t + \delta) - \mathbf{x}_{pndm}(t + \delta)| = O(\delta^3). \tag{22}$$

The intermediate numerical solution $\bar{\mathbf{x}}(t + \delta)$ can be transformed into PNDM by introducing the explicit Adams numerical method Małgorzata & Marciniak (2002). We have:

$$\bar{\mathbf{x}}(t + \delta) = e^{\int_t^{t+\delta} f(\tau)\mathrm{d}\tau}\mathbf{x}(t) + G(t)\frac{1}{\delta}\int_t^{t+\delta}\sum_{i=0}^{3} l_{exp}(i)\boldsymbol{\epsilon}_\theta(t - i\delta) + O(\delta^4)\mathrm{d}\tau$$

$$= e^{\int_t^{t+\delta} f(\tau)\mathrm{d}\tau}\mathbf{x}(t) + G(t)\frac{1}{24}(55\boldsymbol{\epsilon}_\theta(t) - 59\boldsymbol{\epsilon}_\theta(t - \delta) + 37\boldsymbol{\epsilon}_\theta(t - 2\delta) - 9\boldsymbol{\epsilon}_\theta(t - 3\delta)) + O(\delta^4)$$

(23)

where $l_{exp}(i)$ is the Lagrange base derived from the function points $\{t - i\delta, \boldsymbol{\epsilon}_\theta(t - i\delta))\}_{i=0}^3$. Thus, it is easy to know that:

$$|\mathbf{x}_{pndm}(t + \delta) - \bar{\mathbf{x}}(t + \delta)| = O(\delta^4).$$

(24)

If the interpolation function points include the current point $(t + \delta, \epsilon_\delta(t + \delta))$, the integral of Eq. 23 can be rewritten as:

$$\bar{\mathbf{x}}(t + \delta) = e^{\int_t^{t+\delta} f(\tau)\mathrm{d}\tau}\mathbf{x}(t) + G(t)\frac{1}{\delta}\int_t^{t+\delta}\sum_{i=0}^{3} l_{imp}(i)\boldsymbol{\epsilon}_\theta(t - i\delta + \delta) + O(\delta^4)\mathrm{d}\tau$$

$$= e^{\int_t^{t+\delta} f(\tau)\mathrm{d}\tau}\mathbf{x}(t) + G(t)\frac{1}{24}(9\boldsymbol{\epsilon}_\theta(t + \delta) + 19\boldsymbol{\epsilon}_\theta(t) - 5\boldsymbol{\epsilon}_\theta(t - \delta) + \boldsymbol{\epsilon}_\theta(t - 2\delta)) + O(\delta^4),$$

(25)

where $l_{imp}(i)$ is the Lagrange base derived from the function points $\{(t-(i-1)\delta, \boldsymbol{\epsilon}_\theta(t-(i-1)\delta))\}_{i=0}^3$. The fixed coefficients are derived by the implicit Adams numerical method Atkinson et al. (2011). We term the numerical solution derived from ERA-Solver as $\mathbf{x}_{era}(t + \delta)$. It reads:

$$\mathbf{x}_{era}(t + \delta) = e^{\int_t^{t+\delta} f(\tau)\mathrm{d}\tau}\mathbf{x}(t) + G(t)\frac{1}{24}(9\bar{\boldsymbol{\epsilon}}_\theta(t + \delta) + 19\boldsymbol{\epsilon}_\theta(t) - 5\boldsymbol{\epsilon}_\theta(t - \delta) + \boldsymbol{\epsilon}_\theta(t - 2\delta))$$

$$= e^{\int_t^{t+\delta} f(\tau)\mathrm{d}\tau}\mathbf{x}(t) + G(t)\frac{1}{24}(9\boldsymbol{\epsilon}_\theta(t + \delta) + 19\boldsymbol{\epsilon}_\theta(t) - 5\boldsymbol{\epsilon}_\theta(t - \delta) + \boldsymbol{\epsilon}_\theta(t - 2\delta))$$

$$+ G(t)\frac{9}{24}(\bar{\boldsymbol{\epsilon}}_\theta(t + \delta) - \boldsymbol{\epsilon}_\theta(t + \delta)),$$

(26)

where $\bar{\boldsymbol{\epsilon}}_\theta(t + \delta)$ is derived from the $k$-order Lagrange interpolation function which is enhanced with the error-robust selection strategy. Assuming that $G(t)$ is bounded, we can have:

$$|\bar{\mathbf{x}}(t + \delta) - \mathbf{x}_{era}(t + \delta)| = |G(t)|\frac{9}{24}|\bar{\boldsymbol{\epsilon}}_\theta(t + \delta) - \boldsymbol{\epsilon}_\theta(t + \delta)| + O(\delta^4)$$

$$= O(\delta^k) + O(\delta^4).$$

(27)

Finally, we can have:

$$|\mathbf{x}(t + \delta) - \mathbf{x}_{era}(t + \delta)| \leq |\mathbf{x}(t + \delta) - \mathbf{x}_{pndm}(t + \delta)| + |\mathbf{x}_{pndm}(t + \delta) - \bar{\mathbf{x}}(t + \delta)|$$

$$+ |\bar{\mathbf{x}}(t + \delta) - \mathbf{x}_{era}(t + \delta)|$$

$$= O(\delta^3) + O(\delta^4) + O(\delta^k)$$

$$= O(\delta^3) + O(\delta^k)$$

(28)

Finally, we can prove that, when the Lagrange interpolation order $k \geq 3$, the local approximation error of ERA-Solver is $O(\delta^3)$ and has a second-order convergence.

## C.2 PROOF OF THEOREM 2

Firstly, we remove rounded symbols $\lfloor \rfloor$ for ease of analysis. Then, the margin between the adjoin selected indexes can be written as:

$$D_m = i * (\frac{m}{k}^{\lambda(\epsilon)} - \frac{m + 1}{k}^{\lambda(\epsilon)}),$$

(29)

where $\lambda(\epsilon) = \frac{\Delta\epsilon}{\lambda}$. We need to prove that when $\Delta\epsilon$ is large, the indexes are concentrated on the large time, and vice versa. The aggregation can be described by observing the monotonicity of $D_m$. For example, if $D_m < D_{m+1}$, the selected indexes will be gathered at the beginning of the buffer ($t$ is

close to $T$). Through simplification, we just need to analyse the monotonicity of the function $d(m)$ with respect to $m$:

$$d(m) = (m+1)^{\lambda(\epsilon)} - m^{\lambda(\epsilon)}. \tag{30}$$

We can obtain the first order derivative function as follows by taking the derivative:

$$d'(m) = \lambda(\epsilon)((m+1)^{\lambda(\epsilon)-1} - m^{\lambda(\epsilon)-1}). \tag{31}$$

Through observation, we can notice that when $\lambda(\epsilon) > 1$, $d'(m) > 0$, which means the margin function $d(m)$ is monotonically increasing with respect to $m$. Thus the selected indexes are more concentrated on the beginning of buffer, when $\Delta\epsilon$ is large enough ($> \lambda$). Our hyperparameter $\lambda$ is used to control the critical point of concentration.

## D    EXPERIMENTS DETAILS

### D.1    EXPERIMENT SETTINGS

We choose the most widely-used variance preserving (VP) type DPMs Song et al. (2020b) for experiments. We test our method for sampling based on five datasets: Cifar10 ($32 \times 32$) Krizhevsky et al. (2009), LSUN-Church ($256 \times 256$), Yu et al. (2015), LSUN-Bedroom ($256 \times 256$) Yu et al. (2015), CelebA ($64 \times 64$) Liu et al. (2015), and ImageNet ($64 \times 64$) Russakovsky et al. (2014). For pretrained DPMs, we mainly consider continuous DPM Song et al. (2020b), DDIM Song et al. (2020a), Guided Diffusion Dhariwal & Nichol (2021), and Latent Diffusion Rombach et al. (2022) to perform sampling. For all experiments, we evaluate ERA-Solver on NVIDIA V100 GPUs. The computation resources are not limited since the batch size of the sampling can be tuned.

We adopt Frechet Inception Distance (FID) Heusel et al. (2017) as the evaluation metric to test the generation quality of all sampling methods. All evaluation results are based on $50k$ generated samples. Since the network evaluation operation is the main time-consuming operation, the number of function evaluations (NFE) is introduced to align the total sampling time of different fast solvers.

### D.2    SAMPLE QUALITY COMPARISON.

When compared with DEIS Zhang & Chen (2022), the FID results of DEIS in ImageNet $64 \times 64$, Cifar10 (continuous-time DPM), and CelebA are cited from the original paper of DEIS, with the best settings. The other results in LSUN-Church, and LSUN-bedroom are evaluated using the official code integrated into diffusers von Platen et al. (2022).

We directly use the official code released in Lu et al. (2022a) to implement DPM-Solver-fast Lu et al. (2022a) and DPM-Solver++ methods to generate the samples for evaluation. The code license is Apache License 2.0. The other fast solvers like DEIS, and PNDM are implemented based on the official code integrated in diffusers von Platen et al. (2022) if the FID results can not be cited from the original paper. Their sampling setting is the default. Note that the PNDM method integrated in diffusers is a little different from the method Liu et al. (2021)in the paper, which can be seen as the improved version.

The FID results of other methods like Analytic-DDIM in Cifar10 (discrete-time DPM Song et al. (2020a)) and Celeba ((discrete-time DPM Song et al. (2020a))) are directly cited from the paper pf DPM-Solver Lu et al. (2022a). The FID results of other methods like FON in LSUN-church (discrete-time DPM Song et al. (2020a)) are directly cited from the paper pf PNDM Liu et al. (2021).

For the timestep scheme, we use 'linear' and 'logSNR'. 'linear' timestep scheme constructs $\{t_i\}_{i=1}^N$ by uniformly sampling from $[t_N, 1]$. 'logSNR' timestep Lu et al. (2022a) scheme utilizes $\lambda_i = \log \frac{\alpha_i}{\sigma_i}$ as the unit of sampling timesteps and constructs $\{\lambda_i\}_{i=1}^N$ by uniformly sampling from $[\lambda(t_N), 1]$. $t_N$ is the minimum sampling time.

**Cifar10.**    We use the checkpoints of both the discrete-time diffusion models Song et al. (2020a) and continuous-time diffusion models Song et al. (2020b) for comparisons. The comparison results are shown in Tab. 2 and Tab. 3. Following DPM-Solver, we provide results based on two settings of the minimum sampling time $t_N$. The timestep scheme is set to 'logSNR'. On discrete-time diffusion

models, ERA-Solver achieves better generation quality in most NFEs, specifically when NFE is small. On continuous-time diffusion models, the improvement of ERA-Solver is more obvious. ERA-Solver has better generation results on all NFEs. The Lagrange interpolation order $k$ to $4$ and the hyperparameter $\lambda$ are set to $5.0$ for the discrete-time diffusion model and $10$ for the continuous-time diffusion model.

**LSUN-bedroom.** We use the pretrained checkpoint provided by Dhariwal & Nichol (2021) of the discrete-time diffusion model. The comparison results are shown in Tab. 5. The minimum sampling time $t_N$ is set to $1e-4$ for all experiments. The timestep scheme is set to 'linear'. From the table, we can observe that ERA-Solver has better sampling quality on all NFEs. The improvement margin is more obvious when NFE is small. The Lagrange interpolation order $k$ to $4$ and the hyperparameter $\lambda$ is set to $1.0$ for the discrete-time diffusion model.

**LSUN-church.** We use the pretrained checkpoint provided by DDIM Song et al. (2020a) of the discrete-time diffusion model and Latent Diffusion Rombach et al. (2022). The comparison results are shown in Tab. 7 and Tab. 6. The minimum sampling time $t_N$ is set to $1e-4$ for all experiments. The timestep scheme is set to 'linear'. From the two tables, we can observe that ERA-Solver has better sampling quality on most NFEs. The Lagrange interpolation order $k$ is set to $4$ for two pretrained models. The hyperparameter $\lambda$ is set to $5.0$ for the DDIM diffusion model and set to $20.0$ for the Latent Diffusion model.

**CelebA** We use the pretrained checkpoint provided by DDIM Song et al. (2020a) of the discrete-time diffusion model. The comparison results are shown in Tab. 4. The minimum sampling time $t_N$ is set to $1e-4$ for all experiments. The timestep scheme is set to 'logSNR'. From the table, we can observe that ERA-Solver has better sampling quality on most NFEs. The Lagrange interpolation order $k$ to $4$ and the hyperparameter $\lambda$ is set to $30$ for the discrete-time diffusion model.

**ImageNet ($64 \times 64$)** We use the pretrained checkpoint provided by Guided Diffusion Dhariwal & Nichol (2021) of the discrete-time diffusion model. The comparison results are shown in Tab. 8. The minimum sampling time $t_N$ is set to $1e-4$ for all experiments. The timestep scheme is set to 'logSNR'. From the table, we can observe that ERA-Solver has better sampling quality on most NFEs. The Lagrange interpolation order $k$ to $4$ and the hyperparameter $\lambda$ is set to $20.0$ for the discrete-time diffusion model.

Table 2: FID results based on Cifar10 dataset and discrete-time diffusion model Song et al. (2020a).

| Sampling method \ NFE | 8 | 10 | 12 | 15 | 20 | 40 |
|---|---|---|---|---|---|---|
| DDPM Ho et al. (2020) | \ | 278.67 | 246.29 | 197.63 | 137.34 | \ |
| Analytic-DDPM Bao et al. (2022) | \ | 35.03 | 27.69 | 20.82 | 15.35 | \ |
| Analytic-DDIM Bao et al. (2022) | \ | 14.74 | 11.68 | 9.16 | 7.20 | \ |
| DDIM Song et al. (2020a) | 19.23 | 13.58 | 11.02 | 8.92 | 6.94 | 4.92 |
| PNDM Liu et al. (2021) | 9.65 | 8.28 | 6.39 | 5.39 | 4.67 | 3.53 |
| DEIS Zhang & Chen (2022) | 15.35 | 11.35 | 9.53 | 5.56 | 4.60 | 3.60 |
| DPM-Solver++ Lu et al. (2022b) | 14.41 | 10.88 | 9.17 | 5.65 | 4.67 | 3.66 |
| DPM-Solver-fast Lu et al. (2022a) ($t_N = 10^{-3}$) | 28.94 | 6.37 | 4.65 | **3.78** | 4.28 | 3.80 |
| DPM-Solver-fast Lu et al. (2022a) ($t_N = 10^{-4}$) | 80.06 | 11.32 | 7.31 | 4.75 | 3.80 | 3.51 |
| ERA-Solver ($t_N = 10^{-3}, \lambda = 5.0$) | **9.63** | **5.14** | **4.38** | 3.86 | **3.79** | 3.97 |
| ERA-Solver ($t_N = 10^{-4}, \lambda = 5.0$) | 13.01 | 6.16 | 4.84 | 4.2 | 3.84 | **3.45** |

# E  ABLATION ANALYSIS.

In this part, we conduct ablation experiments to demonstrate the effectiveness of the proposed selection strategy of Lagrange bases and the error measure.

Table 3: FID results based on Cifar10 dataset and continuous-time diffusion model Song et al. (2020b).

| Sampling method \ NFE | 8 | 10 | 12 | 15 | 20 | 40 |
|---|---|---|---|---|---|---|
| PNDM Liu et al. (2021) | 9.15 | 7.25 | 6.40 | 5.20 | 4.26 | 2.94 |
| DPM-Solver++ Lu et al. (2022b) | 9.45 | 6.06 | 4.73 | 3.89 | 3.39 | 3.02 |
| DPM-Solver-fast Lu et al. (2022a) ($t_N = 10^{-3}$) | 24.64 | 4.96 | 3.99 | 3.05 | 3.16 | 2.78 |
| DPM-Solver-fast Lu et al. (2022a) ($t_N = 10^{-4}$) | 62.83 | 7.76 | 5.60 | 3.79 | 2.97 | 2.73 |
| DEIS Zhang & Chen (2022) | \ | 4.17 | \ | \ | 3.33 | 2.99 |
| ERA-Solver ($t_N = 10^{-3}, \lambda = 10.0$) | **6.19** | **3.54** | **3.09** | **2.79** | 2.81 | 2.92 |
| ERA-Solver ($t_N = 10^{-4}, \lambda = 10.0$) | 9.98 | 4.46 | 3.48 | 3.02 | **2.76** | **2.67** |

Table 4: FID Results on CelebA Liu et al. (2015) with pretrained discrete-time diffusion model Song et al. (2020a).

| Sampling method \ NFE | 8 | 10 | 12 | 15 | 20 | 40 |
|---|---|---|---|---|---|---|
| DDIM Song et al. (2020a) | 46.44 | 13.4 | 13.23 | 11.63 | 9.62 | 6.87 |
| Analytic-DDPM Bao et al. (2022) | \ | 28.99 | 25.27 | 21.80 | 18.14 | \ |
| Analytic-DDIM Bao et al. (2022) | \ | 15.62 | 13.90 | 12.29 | 10.45 | \ |
| PNDM Liu et al. (2021) | 10.73 | 9.75 | 9.02 | 9.11 | 7.25 | 5.25 |
| DPM-Solver-fast Lu et al. (2022a) | 59.13 | 6.92 | 4.20 | 3.05 | 2.82 | 2.71 |
| DPM-Solver++ Lu et al. (2022b) | 15.64 | 14.16 | 12.76 | 8.36 | 6.49 | 4.52 |
| DEIS Zhang & Chen (2022) | \ | 6.95 | \ | \ | 3.41 | 2.95 |
| ERA-Solver ($\lambda = 30.0$) | **8.76** | **5.06** | **3.67** | **2.99** | **2.75** | **2.69** |

### E.1 EFFECTS OF THE SELECTION STRATEGY

To verify the selection strategy, we replace our error-robust selection strategy (ERS) with the fixed selection strategy (fixed) that fixedly selects the last $k$ estimated noises previously saved in the Lagrange buffer at every sampling step. We select Cifar10 and LSUN-church datasets for evaluation. We use the pretrained checkpoints of discrete-time diffusion models provided by Song et al. (2020a). The results are shown in Tab. 9 and Tab. 10. From the table, we can see that our selection strategy achieves better generation results in general on various Lagrange order $k$ settings. Furthermore, we can obverse that when the Lagrange order $k$ is high, the effect of ERS is more obvious. It demonstrates the potential of the proposed error-robust selection strategy.

We further visualize the comparison results in Fig. 7(b). The results in Fig. 7(b) are achieved based on 5-order Lagrange interpolation methods. High-order interpolation methods may cause the Runge phenomenon Fornberg & Zuev (2007) and bring the oscillation errors of functions. Thus, the top results in Fig. 7(b) exist obvious artifacts. We designed it to better demonstrate the effect of our proposed ERS. It implies that ERS can even be robust to the oscillation errors of the Runge phenomenon partially.

Table 5: FID Results on LSUN-bedroom Yu et al. (2015) with pretrained discrete-time diffusion model Dhariwal & Nichol (2021).

| Sampling method \ NFE | 8 | 10 | 12 | 15 | 20 |
|---|---|---|---|---|---|
| PNDM Liu et al. (2021) | 11.01 | 9.28 | 8.36 | 7.08 | 5.54 |
| DEIS Zhang & Chen (2022) | 11.47 | 9.43 | 8.56 | 6.34 | 5.4 |
| DPM-Solver++ Lu et al. (2022b) | 11.36 | 9.75 | 8.87 | 7.88 | 6.04 |
| UniPC Zhao et al. (2023) | 11.15 | 9.90 | 9.07 | 7.98 | 6.78 |
| ERA-Solver ($\lambda = 1.0$) | **9.09** | **7.28** | **6.93** | **6.25** | **4.9** |

Table 6: FID Results on LSUN-church Yu et al. (2015) with pretrained latent diffusion model Rombach et al. (2022).

| Sampling method \ NFE | 8 | 10 | 12 | 15 | 20 |
|---|---|---|---|---|---|
| PNDM Liu et al. (2021) | 7.09 | 5.29 | 4.87 | 4.37 | 4.45 |
| DPM-Solver++ Lu et al. (2022b) | 7.59 | 5.43 | 4.65 | 7.49 | 6.55 |
| DEIS Zhang & Chen (2022) | 8.89 | 6.23 | 5.13 | 4.68 | 4.75 |
| ERA-Solver ($\lambda$=20.0) | **6.56** | **5.02** | **4.61** | **4.15** | **4.21** |

Table 7: FID Results on LSUN-church Yu et al. (2015) with pretrained discrete-time diffusion model Song et al. (2020a).

| Sampling method \ NFE | 8 | 10 | 12 | 15 | 20 |
|---|---|---|---|---|---|
| DDIM Song et al. (2020a) | 25.40 | 19.62 | 15.77 | 13.31 | 11.75 |
| FON Liu et al. (2021) | \ | \ | \ | 21.32 | 10.3 |
| PNDM Liu et al. (2021) | 14.49 | 11.76 | 10.41 | 9.51 | 9.1 |
| DPM-Solver-2 Lu et al. (2022a) | 238.23 | 23.01 | 16.56 | 13.68 | 11.59 |
| DPM-Solver-fast Lu et al. (2022a) | 140.50 | 19.81 | 13.35 | 11.52 | 10.64 |
| DPM-Solver++ Lu et al. (2022b) | **12.84** | 10.41 | 9.32 | 12.68 | 13.04 |
| DEIS Zhang & Chen (2022) | 14.35 | 11.35 | 9.93 | 10.21 | 10.16 |
| ERA-Solver ($\lambda$=5.0) | 13.15 | **9.42** | **8.15** | **7.41** | **7.39** |

Table 8: FID Results on ImageNet $64 \times 64$ with pretrained Guided Diffusion model Dhariwal & Nichol (2021).

| Sampling method \ NFE | 8 | 10 | 12 | 15 | 20 |
|---|---|---|---|---|---|
| PNDM Liu et al. (2021) | 24.59 | 13.65 | 11.26 | 8.11 | 6.35 |
| DPM-Solver-fast Lu et al. (2022a) | 9.83 | 6.74 | 5.32 | 4.16 | 3.37 |
| DPM-Solver++ Lu et al. (2022b) | 9.60 | 6.18 | 4.89 | 3.87 | 3.23 |
| DEIS Zhang & Chen (2022) | \ | 6.65 | 3.99 | \ | 3.21 |
| ERA-Solver ($\lambda = 20.0$) | **9.30** | **5.11** | **3.91** | **3.24** | **2.89** |

Table 11: Ablation experiments of different prediction types. 'uniform' means the strategy of selecting $k$ Lagrange bases from the buffer with moderate spacing. FID Results are evaluated on Cifar10 Yu et al. (2015) and pretrained continuous-time diffusion model Song et al. (2020b).

| Method\ NFE | | 8 | 10 | 12 | 15 | 20 | 40 |
|---|---|---|---|---|---|---|---|
| | fixed | 12.39 | 6.81 | 4.37 | 3.28 | 2.81 | .62 |
| ERA-Solver ($\epsilon$ prediction) | uniform | 10.18 | 5.77 | 4.85 | 4.27 | 3.54 | 2.94 |
| | ERS | **9.98** | **4.46** | **3.48** | **3.02** | **2.76** | **2.67** |
| ERA-Solver ($x_0$ prediction) | fixed (error-robust) | **15.74** | **7.48** | **3.90** | **3.23** | **2.87** | **2.66** |
| | uniform | 62.71 | 27.16 | 15.39 | 7.85 | 4.74 | 2.98 |

E.2 EFFECTS OF THE ERROR MEASURE

To verify the proposed error measure $\Delta\epsilon$ in the sampling process, we parameterize the power function (Eq. 15 in the main paper) with various constants instead of our error measure. We conduct ablation experiments on Cifar10 and discrete-time checkpoint Song et al. (2020a) and the comparison results are shown in Fig. 7(a).

Table 9: Ablations of ERS on Cifar10

| Method\ NFE | | 10 | 15 | 20 |
|---|---|---|---|---|
| ERA-Solver-3 | fixed | 5.95 | 4.62 | 4.24 |
| | ERS | **5.79** | **4.31** | **4.07** |
| ERA-Solver-4 | fixed | 6.4 | 4.46 | 4.1 |
| | ERS | **5.14** | **3.86** | **3.79** |
| ERA-Solver-5 | fixed | 17.21 | 15.11 | 17.47 |
| | ERS | **6.26** | **3.73** | **3.69** |
| ERA-Solver-6 | fixed | 36.34 | 51.58 | 83.39 |
| | ERS | **19.26** | **4.16** | **3.73** |

Table 10: Ablations of ERS on LSUN-church

| Method\ NFE | | 10 | 15 | 20 |
|---|---|---|---|---|
| ERA-Solver-3 | fixed | **9.83** | 8.52 | 8.72 |
| | ERS | 10.2 | **8.03** | **7.63** |
| ERA-Solver-4 | fixed | 10.56 | 8.72 | 8.99 |
| | ERS | **9.42** | **7.41** | **7.39** |
| ERA-Solver-5 | fixed | 26.7 | 30.36 | 31.58 |
| | ERS | **10.85** | **7.48** | **7.28** |
| ERA-Solver-6 | fixed | 63.91 | 191.69 | 315.6 |
| | ERS | **13.79** | **8.41** | **7.41** |

E.3 EFFECTS ON OTHER PREDICTION TYPES

The motivation and results are mainly based on the noise prediction model. In this part, we explore the performance of ERA-Solver on $x_0$ prediction diffusion model Kingma et al. (2021). The $x_0$ prediction model can be seen as a simple linear transformation of the $\epsilon$ prediction model, which reparameterizes the model prediction as $x_\theta = (x_t - \sigma_t \epsilon_\theta)/\alpha_t$ Kingma et al. (2021).

Firstly, we want to claim that our error-robust selection aims to select the predictions with lower errors when the current error increases. Benny & Wolf (2022) has demonstrated that $x_0$ prediction diffusion model has completely opposite error changing trends compared with $\epsilon$ prediction diffusion models. When sampling time $t$ approaches 0, the error of $x_0$ prediction will become small. That means, the error-robust strategy for the $x_0$ prediction diffusion model turns out to be the fixed selection strategy, which fixedly selects the last $k$ predictions at every sampling step since the predictions at the end of the buffer have already owned the lowest errors.

We conduct ablation experiments on Cifar10 and continuous-time pretrained diffusion models. We further introduce the uniform selection strategy ($\frac{\Delta\epsilon}{\lambda}$ of Eq. 17 in the main paper is changed to be 1), which uniformly selects $k$ estimated noises previously saved in the Lagrange buffer at every sampling step, for reference. The results are shown in Tab. 11. From the table, we can observe that the uniform selection strategy performs worse than the fixed selection strategy when the prediction type is $x_0$ and performs better than the fixed selection strategy when the prediction type is $\epsilon$. This phenomenon indicates that different error trends have different optimal selection strategies.

We can also obverse that ERA-Solver based on the noise prediction model achieves better sampling quality than that based on the data prediction model, which means that our designed error-robust selection strategy for the noise prediction model is still worthy.

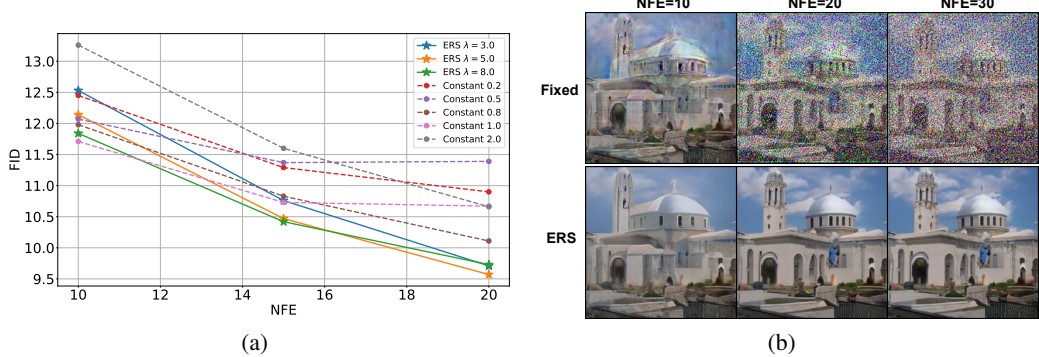

(a)               (b)

Figure 7: (**left**) Ablation results of error-aware scale ($\Delta\epsilon/\lambda$ in Eq. 15) and constant scale (replace $\Delta\epsilon/\lambda$ with a constant) based on the 3-order Lagrange interpolation function and 5000 generated samples. (**right**) Ablated visualization results of fixed selection strategy and error-robust selection strategy.

Table 12: Computation time per sample of sampling on Guided Diffusion Dhariwal & Nichol (2021), varying different solvers and NFE.

| Sampling Method \ NFE | 8 | 10 | 12 | 15 | 20 |
|---|---|---|---|---|---|
| PNDM Liu et al. (2021) | 0.388 | 0.482 | 0.581 | 0.723 | 0.966 |
| DPM-Solver++ Lu et al. (2022a) | 0.401 | 0.490 | 0.592 | 0.738 | 0.980 |
| DEIS Zhang & Chen (2022) | 0.396 | 0.488 | 0.589 | 0.734 | 0.978 |
| ERA-Solver | 0.413 | 0.512 | 0.617 | 0.792 | 1.062 |

### E.4 TECHNOLOGY LIMITATION

In this part, we describe the limitation of ERA-Solver. Since Error-Robust Selection strategy tends to take all previous estimated noises into consideration, the maintained buffer will be longer. Thus, the computation time will be slightly more than other methods. We provide the computation time per sample in the Table. 12.

In practical scenarios like Stable Diffusion, ERA-Solver can already generate realistic samples when NFE is around 20 (Fig. 11, Fig. 9, and Fig. 10). Thus, the negative impact of computing time and memory cost can be limited.

Although there exists limitation, ERA-Solver is ensured to bring sampling quality improvement. For example, in Table. 3 and Table. 5, ERA-Solver with NFE 10 can outperform previous methods with NFE 12 (3.54 vs 3.99, 7.28 vs 8.36), while the computing time of ERA-solver with NFE 10 is lower than previous methods with NFE 12 (0.512 vs 0.581).

## F QUALITATIVE RESULTS

### F.1 RESULTS ON UNCONDITIONAL DIFFUSION MODEL

We sample and visualize the generated samples from the discrete-time diffusion model Dhariwal & Nichol (2021) pretrained on LSUN-church. We select PNDM Liu et al. (2021), DPM-Solver++ Lu et al. (2022b), and DEIS Zhang & Chen (2022) to compare with ERA-Solver. For ERA-Solver, the Lagrange order $k$ is set to $4$, and hyperparameter $\lambda$ is set to $1$. We align the sampling NFE and the random seed for a fair comparison. The comparison results are shown in Fig. 8. From Fig. 8, we can observe that ERA-Solver can generate more natural textures than other fast solvers, specifically when NFE is small.

## F.2 RESULTS ON THE TEXT-TO-IMAGE DIFFUSION MODEL

In this part, we sample from the large-scale latent diffusion model, i.e., Stable Diffusion Rombach et al. (2022) with different fast solvers and ERA-Solver. We select PNDM and DPM-Solver++ for comparison, the codes of which are all applied from diffusers von Platen et al. (2022). For ERA-Solver, we set $k = 4$ and $\lambda = 10.0$. It can be observed that ERA-Solver can generate promising images when NFE is 15, which is faster than DPM-Solver++ and PNDM. It demonstrates that ERA-Solver can be extended to various generative applications and has the potential to promote the progress of the art creation industry. The comparison results are shown in Fig. 9, Fig. 10, and Fig. 11.

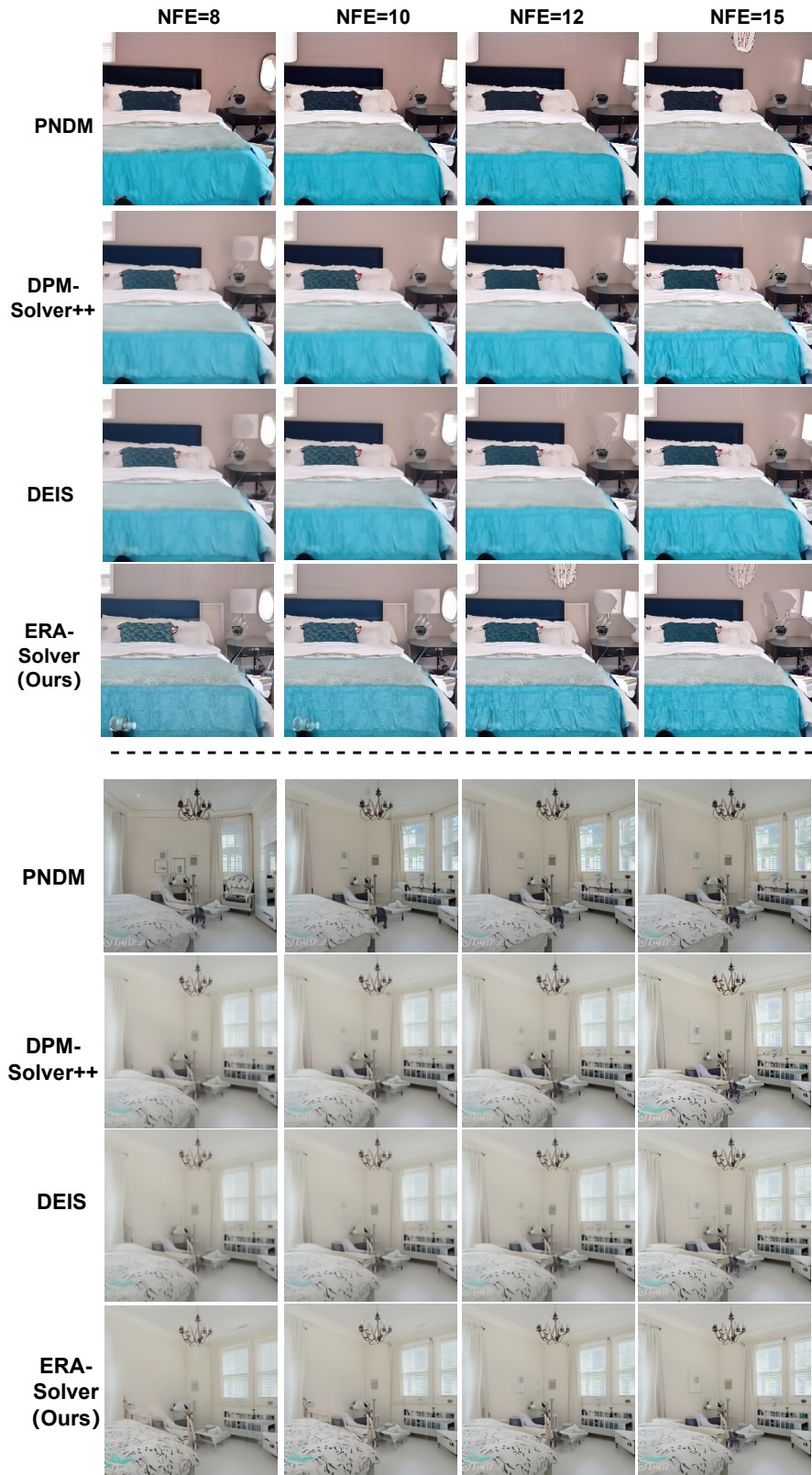

Figure 8: Generation quality comparison based on 5, 8, and 10 NFEs. The error of estimated noises tends to appear at $t_i$ close to $0$ with high-frequency information generated. Our ERA-Solver is robust to the error so as to generate natural textures while DPM-Solver-fast fails.

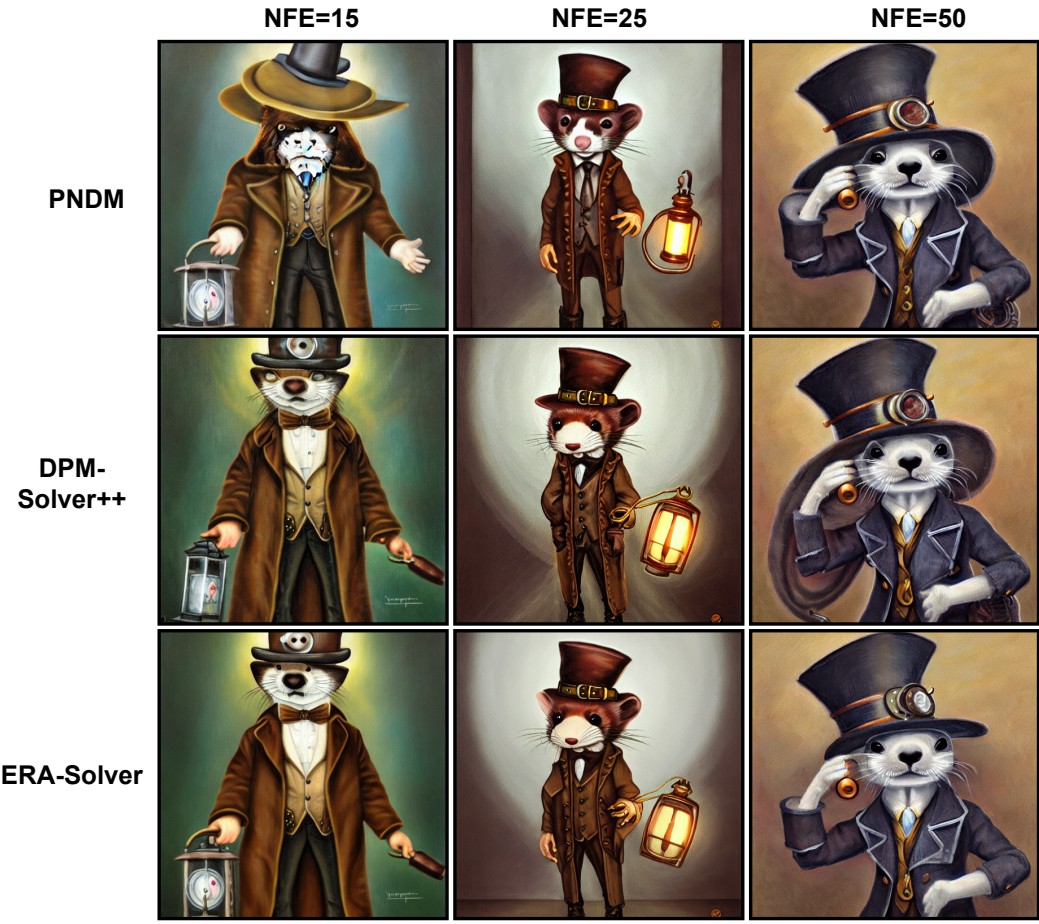

Figure 9: Samples using the pretrained Stable-Diffusion Rombach et al. (2022) with a classifier-free guidance scale 7.5 (the default setting), varying different solvers and NFEs. The main part of the input prompt is: "Cute and adorable ferret wizard, wearing coat and suit".

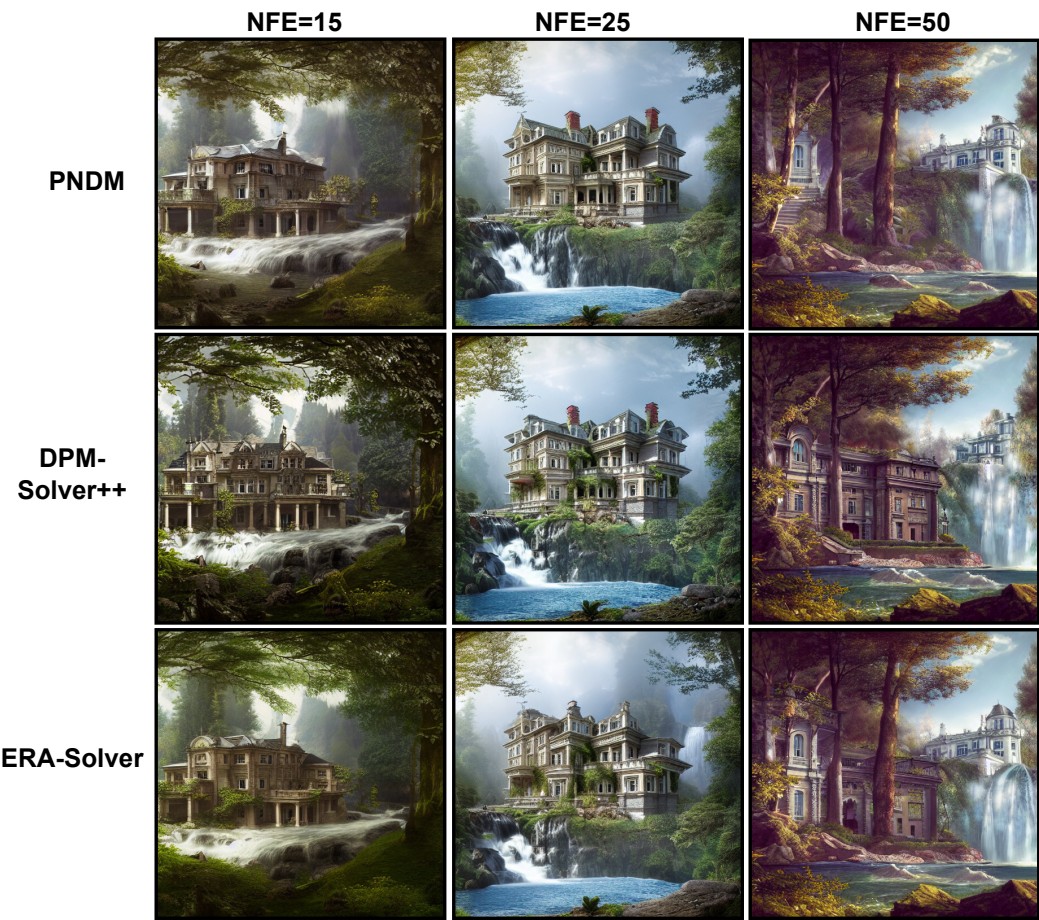

Figure 10: Samples using the pretrained Stable-Diffusion Rombach et al. (2022) with a classifier-free guidance scale 7.5 (the default setting), varying different solvers and NFEs. The main part of the input prompt is: "A beautiful mansion beside a waterfall in the woods".

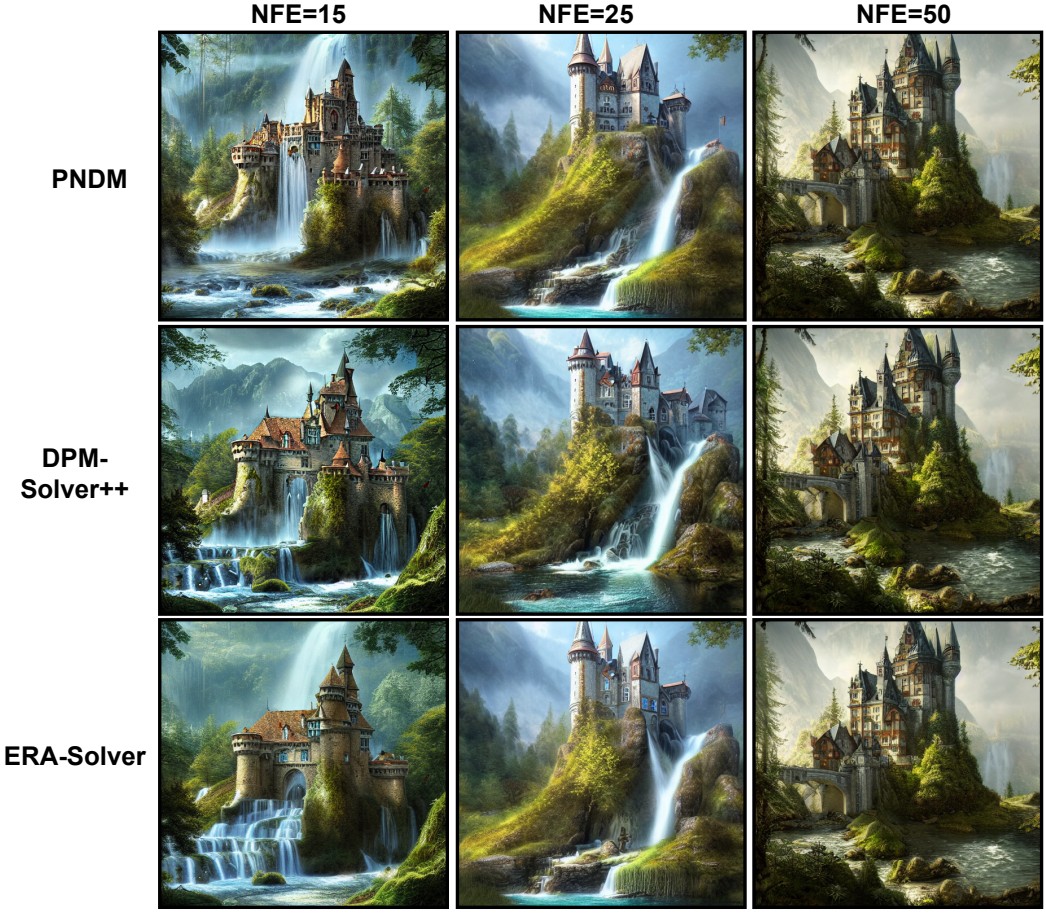

Figure 11: Samples using the pretrained Stable-Diffusion Rombach et al. (2022) with a classifier-free guidance scale 7.5 (the default setting), varying different solvers and NFEs. The main part of the input prompt is: "A beautiful castle beside a waterfall in the woods".

