# OpenReview forum: "ERA-Solver: Error-Robust Adams Solver for Fast Sampling of Diffusion Probabilistic Models"
_ICLR.cc/2024/Conference — Submitted to ICLR 2024_

### Official Review · Reviewer_L8Et · 2023-10-24

**Soundness:** 3 good
**Presentation:** 2 fair
**Contribution:** 2 fair
**Rating:** 5
**Confidence:** 3

**Summary:**

The paper proposes a strategy to select a fixed number of estimated Gaussian noises from the buffer per timestep t_i, and then used the selected ones in estimating the next diffusion state via the implicit Adams numerical method.  The selection strategy intends to minimize the prediction error of the estimated Gaussian noises. Different experiments are performed, showing the effectiveness of the new sampling method.

**Strengths:**

The paper proposes a new method for selecting a fixed number of the estimated Gaussian noises from the buffer per timestep to better compute the next diffusion state. It seems that the search procedure is performed online for each individual sampling, which is interesting.

**Weaknesses:**

(1). The method needs to introduce a buffer to store all the historical estimated Gaussian noises up to the most recent timestep. As the timestep t_i approaches to 0, the buffer size is increasingly large which is undesirable from a practical point of view.

(2). As the buffer becomes large along with the timestep, one can imagine that it would take more time to do searching. Therefore, the method requires not only more memory but also more sampling time.

**Questions:**

(1) I think Theorem 2 is not properly formulated because the term "large enough" cannot be quantified.

(2) I don't get which line in Algorithm 1 performs selection of the estimated Gaussian noises from the buffer.

(3) It is not clear from the paper how the search is performed. Is it greedy search?

---

### Official Review · Reviewer_hCzL · 2023-10-31

**Soundness:** 3 good
**Presentation:** 3 good
**Contribution:** 2 fair
**Rating:** 5
**Confidence:** 4

**Summary:**

This paper proposes an error-robust Adams solver (ERA-Solver) that consists of a predictor (similar to the Adams-Bashforth method) and corrector (similar to the Adams-Moulton method). The authors propose an error-robust selection strategy to select the former evaluations rather than the last $k$ evaluations in the predictor. The experiment result shows it can achieve good sample qualities at a few NFEs.

**Strengths:**

$\cdot$ The writing of this paper is clear and easy to follow.

$\cdot$ The authors take the score estimation error into consideration in the sampling process, which is novel. The authors conduct some simple experiments to verify that as $t\rightarrow 0$, the error in terms of $\epsilon$ becomes larger.

**Weaknesses:**

$\cdot$ Some experiment comparison results are unfair. For example, in Table 5 in this paper, on the LSUN-bedroom dataset, the DPM-Solver++[2] achieves 6.04 FID in 20 NFEs. However, in the DPM-Solver[1] paper, DPM-Solver achieves 3.09($\epsilon = 1e-3$)/2.60($\epsilon = 1e-4$) FID in 20 NFEs. This paper and [1] both use the pretrained diffusion models offered in [3]. I guess the reason is that in this paper the authors use the 'time linear space' schedule for timesteps while [1] use the 'logSNR linear space' schedule. I suggest the authors should compare these methods in a better setting.



[1] DPM-Solver: A Fast ODE Solver for Diffusion Probabilistic Model Sampling in Around 10 Steps, Lu et al.

[2] DPM-Solver++: Fast Solver for Guided Sampling of Diffusion Probabilistic Models, Lu et al.

[3] Diffusion Models Beat GANs on Image Synthesis, Dhariwal et al.

**Questions:**

See weaknesses

---

### Official Review · Reviewer_khj4 · 2023-10-31

**Soundness:** 3 good
**Presentation:** 1 poor
**Contribution:** 3 good
**Rating:** 3
**Confidence:** 3

**Summary:**

Some fast sampling methods of DDPMs rely on the equivalence between sampling and solving an ODE on the noise process. They then leverage various ODE solvers. The main remark of the paper is to notice a significant discrepancy between the theoretical noise and the noise practically estimated by the trained diffusion model. This noise is also specific to each dataset. The authors hence propose a technical solution to include this uncertainty in the ODE solver and show numerically how their approach outperforms existing approaches on several classical datasets.

**Strengths:**

The remarks and technical solutions of the authors for accelerating ODE-based fast sampling DDPMs are novel and well-conducted.

**Weaknesses:**

There are three weaknesses in this paper:
- The motivation of the paper is not compelling. The argument is that the main drawback of DDPMs is the sampling time and that there are only two areas of research: 1) fast ODE samplers and 2) distillation or learning-based samplers. The authors need to mention latent diffusion models, which are the go-to solution for fast sampling for the generating tasks mentioned in the paper, i.e., classical natural image problems. How does their approach even compare to latent diffusion models?
For the authors' argument to be compelling, I suggest they experiment with their approach to generating tasks where latent diffusion models are complex to leverage, i.e., in tasks where we need access to a good latent representation of the data. Alternatively, it would be interesting to see how their approach can be used to accelerate the sampling of latent diffusion models further. Because otherwise, the paper is interesting as a new technical solution but will not be helpful in practice.
- There are plenty of typing errors and sentences that need meaning. With all the corrector error systems, this is not very pleasant to read. For instance, in the introduction, there are almost systematically missing spaces between words and citations. Alternatively, even a sentence in the abstract that is hardly understandable "are not able to robust with the various error patterns in the noise estimated ...".
- The numerical experiment only showcases FID scores; why not use other scores, such as improved recall and precision?

**Questions:**

None

---

### Meta-Review · Area_Chair_4MNK · 2023-11-30

**Metareview:**

This article makes an interesting remark that existing fast sampling methods for diffusion model are not robust with respect to various error patterns in the noise. A new integrator is proposed to address this issue. However, reviewers unanimously raised several concerns regarding both the presentation and the sufficiency of demonstration. Therefore, I cannot recommend acceptance, but encourage the authors to consider providing further evidence in a future submission.

**Justification For Why Not Higher Score:**

I agree with reviewers' assessment.

**Justification For Why Not Lower Score:**

I agree with reviewers' assessment.

---

### Decision · Program_Chairs · 2024-01-16

Reject